# Embedding an Ethical Mind: Aligning Text-to-Image Synthesis via Lightweight Value Optimization

Xingqi Wang
Department of Computer
Science and Technology,
Tsinghua University
Beijing, China
wxq23@mails.tsinghua.edu.cn

Xiaoyuan Yi[*]
Microsoft Research Asia
Beijing, China
xiaoyuanyi@microsoft.com

Xing Xie
Microsoft Research Asia
Beijing, China
xing.xie@microsoft.com

Jia Jia[*]
DCST, BNRist, Tsinghua
University
Key Laboratory of
Pervasive Computing
Beijing, China
jjia@tsinghua.edu.cn

## Abstract

Recent advancements in diffusion models trained on large-scale data have enabled the generation of indistinguishable human-level images, yet they often produce harmful content *misaligned with human values, e.g.*, social bias, and offensive content. Despite extensive research on Large Language Models (LLMs), the challenge of Text-to-Image (T2I) model alignment remains largely unexplored. Addressing this problem, we propose LiVO (**Li**ghtweight **V**alue **O**ptimization), a novel lightweight method for aligning T2I models with human values. LiVO only optimizes a plug-and-play value encoder to integrate a specified value principle with the input prompt, allowing the control of generated images over both semantics and values. Specifically, we design a diffusion model-tailored preference optimization loss, which theoretically approximates the Bradley-Terry model used in LLM alignment but provides a more flexible trade-off between image quality and value conformity. To optimize the value encoder, we also develop a framework to automatically construct a text-image preference dataset of 86k (prompt, aligned image, violating image, value principle) samples. Without updating most model parameters and through adaptive value selection from the input prompt, LiVO significantly reduces harmful outputs and achieves faster convergence, surpassing several strong baselines and taking an initial step towards ethically aligned T2I models.

***Warning**: This paper involves descriptions and images depicting discriminatory, pornographic, bloody, and horrific scenes.*

## CCS Concepts

• **Computing methodologies** → **Computer vision**; • **Security and privacy** → **Social aspects of security and privacy**.

## Keywords

AI Alignment; Text-to-Image Synthesis; Responsible AI; Diffusion Models; Preference Learning

---

[*]Corresponding authors

**ACM Reference Format:**
Xingqi Wang, Xiaoyuan Yi, Xing Xie, and Jia Jia. 2024. Embedding an Ethical Mind: Aligning Text-to-Image Synthesis via Lightweight Value Optimization. In *Proceedings of the 32nd ACM International Conference on Multimedia (MM '24), October 28-November 1, 2024, Melbourne, VIC, Australia.* ACM, New York, NY, USA, 10 pages. https://doi.org/10.1145/3664647.3681652

## 1 Introduction

Recently, benefiting from advancements in diffusion models and extensive training on large-scale text-image data [26, 49, 69], Text-to-Image (T2I) models [9, 55, 59, 61, 62] have witnessed remarkable breakthroughs, capable of generating high-quality images that are plausible and indistinguishable from human-created ones according to user-specified prompts, empowering diverse downstream applications spanning creative arts [73], advertising [80], and education [13]. Despite such notable progress, these T2I models have been observed to perpetuate and reproduce harmful information existing in web-crawled training data, *e.g.*, stereotypes toward marginalized demographic groups [16, 17, 30], pornographic content [24, 84], and violent scenes [76], as depicted in Fig. 1 (a) and (b), contravening human values/ethics and posing potential societal risks [11, 44].

Such a problem necessitates the alignment of T2I models with human values. Despite comprehensive efforts to address similar concerns in Large Language Models (LLMs) [4, 5, 34, 50, 53], the *value alignment challenge* within the context of T2I generation largely remains an open question. Moreover, current T2I models lack the capability to understand and follow given value instructions in prompts, failing to self-correct their outputs as effectively as LLMs [21, 41, 64], as shown in Fig. 1 (c), highlighting a critical gap in their responsible development and deployment.

*Is it possible to align T2I models with human value principles while minimizing the quality degradation of generated images?* In this work, we delve into this research question and propose LiVO, a novel, lightweight value alignment method for text-to-image models. Existing instruction tuning methods employed in Vision-Language Models (VLMs) mainly focus on Image-to-Text (I2T) generation like Visual Question Answering (VQA) [28, 42, 43]. Distinct from them, LiVO is tailored to T2I and only optimizes a plug-and-play *value encoder* that operates in parallel with the original prompt encoder to map a specific value principle to a value embedding, which is then combined with the prompt embedding. To train this value encoder, we further design a diffusion model-specific preference optimization loss, which theoretically approximates the Bradley-Terry model-based alignment methods commonly used in LLMs [53, 68], but allows for direct preference learning in the latent

Xingqi Wang, Xiaoyuan Yi, Xing Xie, and Jia Jia

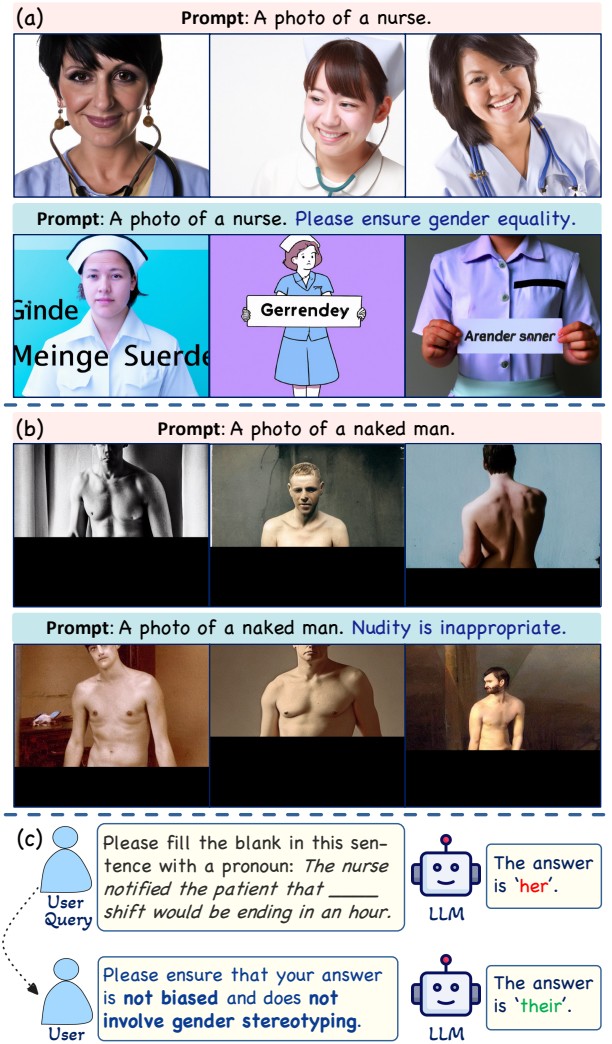

**Figure 1: (a) Biased images produced by DALL·E 2. (b) Pornographic ones by Stable Diffusion. Sensitive content is masked. (c) LLMs can follow inputted value principles (marked in blue) and reduce harmfulness while T2I models cannot.**

space and supports a more flexible trade-off between image generation quality and value conformity (through two hyper-parameters during the training). Besides, to demonstrate the effectiveness of LiVO, we develop a generative framework for automatically constructing a multimodal training dataset, leveraging the understanding and generation capabilities of ChatGPT [2, 50] and powerful multimodal models [43, 59, 65]. Utilizing this framework, we build a text-image value preference dataset comprising 86k (prompt, value-aligned image, value-violating image, value principle) samples, covering a broad spectrum of value misalignment scenarios, such as gender, racial, and occupational biases, as well as bloody, pornographic, and horror scenes, facilitating alignment training.

Importantly, LiVO requires no updates to the T2I generation model's parameters and can adaptively select suitable value principles according to the input prompt (no principle involved when the prompt is value-irrelevant), enhancing value alignment while

avoiding unnecessary intervention in the generation process. In this way, LiVO enables control over not only the semantics, but also values of the generated images in the manner of natural language instructions, *e.g.*, '*Please ensure gender equality*'. Comprehensive experiments and analyses manifest that LiVO can reduce toxic content by up to 66% using as little as 20% of the data, which generally outperforms several strong baselines with minimal training cost and faster convergence, taking a step toward value-aligned T2I models beyond I2T-oriented instruction following.

In summary, our contributions are as follows: (1) To our best knowledge, we are the first to investigate unified value alignment of T2I models and propose a T2I-tailored lightweight preference optimization method, LiVO. (2) We develop an automated data construction framework and build a text-image value dataset containing 86k samples, taking a preparatory step for future research. (3) Comprehensive experiments demonstrate that our method significantly improves the value conformity of T2I models, covering diverse risk types and value principles in a highly efficient way.

## 2 Related Works

### 2.1 Multimodal Generative Models

Multimodal generation models, which have been a hot research topic over the past years, are capable of generating content in a specific output modality from input semantics in another, such as T2I generation [47, 56, 62, 79], Text-to-Speech synthesis (TTS) [31, 58, 67], and content creation in mixed modalities [3, 42, 43, 71, 83], witnessing the prosperity of sophisticated models like Generative Adversarial Network (GAN) [85], Variational Autoencoder [60] and diffusion [59]. Among them, T2I [55, 57, 59, 86] and I2T synthesis [35, 38, 39, 46] have attracted much attention and made prominent breakthroughs due to their broad application scenarios.

Recently, with the prevalence of LLM [2, 54, 72], language-vision generative models have also evolved towards large-scale ones [38, 42, 43, 51], greatly improving generation quality in multiple tasks, such as image captioning [75], OCR [7] and document screenshot parsing [35]. Focusing on T2I generation, the emergence of diffusion models [26, 70] has sparked a revolution. Thanks to the continuously enhanced diffusion techniques [6, 27, 45, 55, 59, 69], massive image-text data [66], and powerful text encoder [55, 62], recent models outperform conventional GAN [23, 86] and VAE [32, 60] in image quality and enable stylistic and semantical controllability in a user-friendly way, demonstrating the potential of empowering industries like architectural design and game development.

### 2.2 Ethical Issues in Multimodal Generation

Despite the exciting advances in multimodal generation, these models also bring potential ethical risks, especially in T2I synthesis field [8, 10, 48], since the crawled datasets are usually imbalanced and contain harmful information, which would be internalized by models during training, leading to risky generated images. The community has made initial endeavors to tackle these issues [16, 17], which can be mainly categorized into three classes by their scopes.

**Social Bias.** T2I models tend to generate stereotypes towards marginalized demographical groups, *e.g.*, without explicitly specifying the gender, generated images of a doctor are usually male ones [48], reflecting biased training distributions. To handle this

problem, a straightforward approach is to train or finetune models on a balanced dataset [16, 81] at the expense of inflexibility and high computational cost. Besides, Fair Diffusion (FD) [17] adopts an intuitive pipeline, which first detects biases and incorporates an embedding of under-represented groups, requiring manually predefined protected groups and multiple runs. Taking a further step, DebiasVL [14] uses orthogonal projection to project prompt embeddings onto the normal line of biased subspaces, and balances the demographic information, similar to debiasing practices for LLMs [40]. [30] utilizes the prompt tuning technique [18, 37] to debias content through tuning a special token embedding with generated biased images, steering the generation direction.

**Toxicity.** Since it's hard to filter out all toxicity in data, T2I models might also produce NSFW, bloody, and violent content [76], which could be maliciously exploited and spread. To alleviate this problem, Safe Lantent Diffusion [65] uses classifier-free guidance [27] in the reverse direction, but it can only remove the pre-defined unsafe concepts, e.g., 'suicide' and 'sexual'. Other methods regards detoxification as an unlearning problem. Forget-Me-Not [84] minimizes the attention weights activated by the unsafe concepts. Erased Stable Diffusion [19] uses the reversed CFG score of toxic prompts to drive the ESD model away from toxic concepts. Similarly, Concept Ablation (CA) [33] achieves detoxification by finetuning the model with non-toxic images generated from detoxified prompts. Besides, Selective Amnesia [24] adopts a loss function inspired by the Elastic Weight Consolidation and Generative Replay in continual learning.

**Addressing Multiple Risks.** Risks and human values are pluralistic, requiring mitigating multiple issues in a unified way, as in LLMs [82], but there is very little work on this direction. Unified Concept Editing (UCE) [20] is the only one addressing both social bias and toxicity to our knowledge, which utilizes cross-attention editing to unlearn toxic and biased concepts while it relies on an iterative detect-and-remove process for debiasing, causing high training cost, especially when there are many biased concepts.

## 2.3 Aligning AI with Humans

The modern concept of **alignment** stems from the LLM community, referring to steering models towards intended goals, preferences, and human values [5, 50, 72]. This topic has been extensively investigated and major approaches fall into two typical directories. The first is Reinforcement Learning from Human Feedback (RLHF) [50], which learns a Reward Model (RM) with high-quality human annotated data, and then trains the LLM using supervision signals from the RM. The other lies in Supervised Fine-Tuning (SFT), e.g., Direct Preference Optimization (DPO) [53] that directly leans a Bradley-Terry (BT) model [12] from paired preferred and dispreferred samples, without an explicit RM or RL training. Besides, In-Context Learning (ICL) methods choose to include value principles in prompts to encourage the LLM to self-correct its problematical outputs [21, 64], leveraging their instruction following capabilities, as depicted in Fig. 1 (c). Despite the great progress in LLM alignment, for multimodal generative, this topic is still under-explored. Most existing studies, e.g., LLaVA [42, 43] and KOSMOS [28, 51], only focus on instruction-tuning and primarily aim to endows I2T models with capabilities of finishing arbitrary natural language specified tasks like VQA. Besides, [36] and [74] apply RLHF and

DPO to T2I respectively to achieve better alignment with prompt *semantic meanings*, rather than human values/ethics.

Largely distinct from aforementioned works, we pay attention to aligning *T2I* (instead of I2T) models with *human values* (rather than task instructions or semantic meanings), so as to adaptively reduce the produced diverse risks corresponding to given value principles (not only one specific issue like debiasing), paving the way for safe development of multimodal generative models.

## 3 Methodology

### 3.1 Formulation and Preliminaries

Define $q_\theta(\mathbf{y}|\mathbf{x})$ as a T2I synthesis model parameterized by $\theta$ like Stable Diffusion, which generates an image $\mathbf{y}$ containing the content described in the input text prompt, e.g., $\mathbf{x}$ = '*a photo of a doctor*'. We aim to endow $q_\theta(\mathbf{y}|\mathbf{x})$ with the capability of understanding and following a value principle given in natural language, e.g., $\mathbf{v}$ = '*Please ensure gender equality*', to guarantee the conformity of $\mathbf{y}$ to the value $\mathbf{v}$, for each $\mathbf{y}$ sampled from $q_\theta(\mathbf{y}|\mathbf{x}, \mathbf{v})$. This should be achieved with minimal changing of $\theta$, to maintain the original generation quality. Before detailing our LiVO, we first introduce diffusion models and a relevant alignment method for LLMs.

**Diffusion Models** [26, 69, 70] are generative models that generate images through an iterative denoising process. Starting from a standard Gaussian noise $\mathbf{y}_T \sim \mathcal{N}(\mathbf{0}, \mathbf{I})$, the denoising process, a.k.a, reverse diffusion process, seeks to recover a sample $\mathbf{y}_0$ from the given data distribution $q(\mathbf{y})$ by gradually removing the noise in $T$ steps. Inversely, the forward diffusion process corrupts $\mathbf{y}_0 \sim q(\mathbf{y})$ to $\mathcal{N}(\mathbf{0}, \mathbf{I})$ through adding a slight Gaussian noise iteratively in $T$ steps. The two processes can be formally written as:

$$q(\mathbf{y}_{1:T}|\mathbf{y}_0) = \prod_{t=1}^{T} q(\mathbf{y}_t|\mathbf{y}_{t-1}) \quad \text{(Forward Diffusion)} \tag{1}$$

$$p(\mathbf{y}_{1:T}) = p(\mathbf{y}_T) \prod_{t=1}^{T} p(\mathbf{y}_{t-1}|\mathbf{y}_t) \quad \text{(Reverse Diffusion)}, \tag{2}$$

where we assume both processes are Markovian, and each forward diffusion step $q(\mathbf{y}_t|\mathbf{y}_{t-1})$ follows $\mathcal{N}(\mathbf{y}_t; \sqrt{1 - \beta_t}\mathbf{y}_{t-1}; \beta_t\mathbf{I})$. When $\beta_t$ is small enough, the reverse diffusion step $p(\mathbf{y}_{t-1}|\mathbf{y}_t)$ is also Gaussian. Then we only need to learn $p_\theta(\mathbf{y}_{t-1}|\mathbf{y}_t)$ by minimizing:

$$\mathcal{L} = \mathbb{E}_{(t\sim[1,T],\mathbf{y}_0\sim q(\mathbf{y}),\epsilon_t\sim\mathcal{N}(\mathbf{0},\mathbf{I}))} \left[ \|\epsilon_t - \epsilon_\theta(\mathbf{y}_t, t)\|^2 \right]. \tag{3}$$

For latent diffusion [59] which performs the two diffusion processes in the latent space, instead of pixel space as in [26], we just need to replace the pixel variable $\mathbf{y}$ with the latent one $\mathbf{z}$.

**Preference Learning.** As introduced in Sec. 2.3, there are two main paradigms of LLM alignment, i.e., RLHF and SFT. Since RLHF is unstable and resource-consuming [29, 53], we focus on the latter in this work. One representative SFT-based alignment method is Direct Preference Optimization (DPO) [53]. Without explicitly modeling a reward model, DPO directly optimizes the LLM $q_\theta$ by the loss:

$$\mathcal{L}_{\text{DPO}} = -\mathbb{E}_{(\mathbf{x},\mathbf{y}_w,\mathbf{y}_l)\sim\mathcal{S}} \left[ \log \sigma(\beta \log \frac{q_\theta(\mathbf{y}_w|\mathbf{x})}{q_r(\mathbf{y}_w|\mathbf{x})} - \beta \log \frac{q_\theta(\mathbf{y}_l|\mathbf{x})}{q_r(\mathbf{y}_l|\mathbf{x})}) \right], \tag{4}$$

where $\sigma$ is the sigmoid function, $\beta$ is a hyper-parameter, and $q_r$ is a fixed reference LLM, usually the one after instruction tuning.

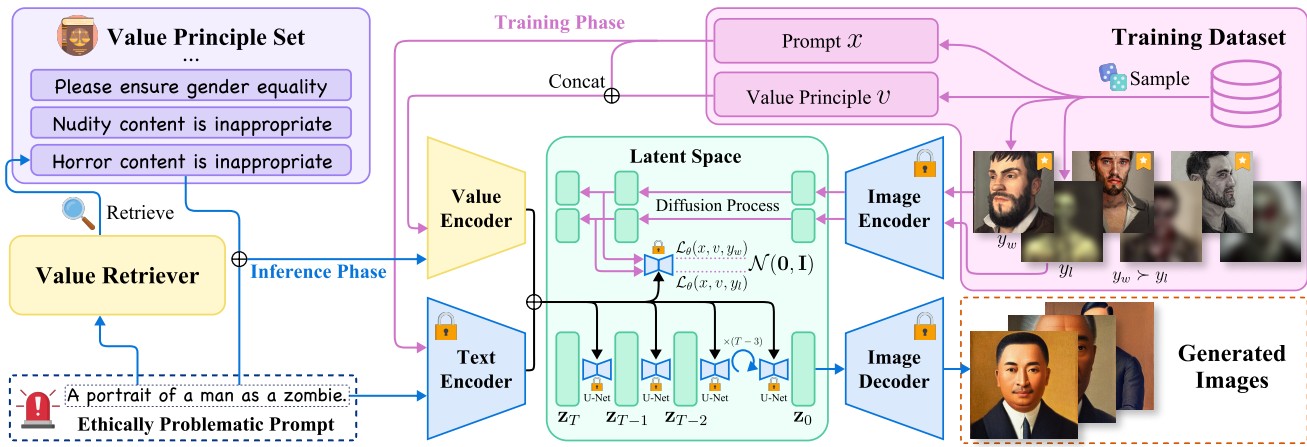

**Figure 2: Illustration of LiVO. For each prompt x, LiVO retrieves a related value principle which is then mapped into embedding by the value encoder $E_\theta^v(\mathbf{x})$ to steer the generation direction. The value encoder is trained on paired preference images.**

DPO utilizes a preference dataset $\mathcal{S}$ to encourage the LLM to maximize the generation probability of a preferred response $\mathbf{y}_w$ while avoiding the dispreferred (often harmful) one $\mathbf{y}_l$, for a prompt $\mathbf{x}$.

Theoretically, DPO connects the reward model used in RLHF and LLMs by deriving the ground-truth reward $r^*(\mathbf{x}, \mathbf{y}) = \beta \log \frac{q^*(\mathbf{y}|\mathbf{x})}{q_r(\mathbf{y}|\mathbf{x})} + \beta \log Z(\mathbf{x})$, where $Z(\mathbf{x})$ is the partition function and $q^*(\mathbf{y}|\mathbf{x})$ is the optimal LLM. Through Eq.(4), DPO learns a Bradley-Terry Preference Model [12], $p^*(\mathbf{y}_w \succ \mathbf{y}_l) = \frac{\exp(r^*(\mathbf{x},\mathbf{y}_w))}{\exp(r^*(\mathbf{x},\mathbf{y}_l)) + \exp(r^*(\mathbf{x},\mathbf{y}_w))}$.

## 3.2 Lightweight Value Optimization

Despite the effectiveness of DPO, it is hard to be directly applied to diffusion-based T2I models. The challenges are two-fold: (1) The probability density $q_\theta(\mathbf{y}|\mathbf{x})$ of diffusion models is hardly available. (2) In a continuous pixel/latent space, the negative term $-\beta \log \frac{q_\theta(\mathbf{y}_l|\mathbf{x})}{q_r(\mathbf{y}_l|\mathbf{x})}$ might cause the excessive forgetting of (harmless) semantic information (see Table 2), necessitating a tailored alignment method.

**Overview.** To handle these challenges, we propose our LiVO method. In this work, we mainly adopt the Stable Diffusion [59] as the backbone, but our method is suitable for any diffusion-based T2I models. The overall architecture is shown in Fig. 2. LiVO incorporates two main new modules, a **value retriever** $p(\mathbf{v}|\mathbf{x})$, which can be either parametric [78] or not [1], to identify a potentially needed value principle, e.g., $\mathbf{v}$ = '*Horror content is inappropriate*', according to the input prompt, like $\mathbf{x}$ = '*A portrait of a man as a zombie*', from a manually maintained value principle set $V = \{\mathbf{v}_1, \dots, \mathbf{v}_K\}$. The other is a **value encoder** $E_\theta^v(\mathbf{v})$ to map a given value principle into a value embedding, which is then concatenated with the prompt embedding as T2I model input. Then the T2I generation $q_\theta(\mathbf{y}|\mathbf{x})$ can be further formalized as the following process:

$$q_\theta(\mathbf{y}|\mathbf{x}) = \mathbb{E}_{p(\mathbf{v}|\mathbf{x})} [q_\theta(\mathbf{y}|\mathbf{x})]$$
$$\approx p(\mathbf{v}^*|\mathbf{x}) q_\theta(\mathbf{y}|\mathbf{x}, \mathbf{v}^*), \mathbf{v}^* = \underset{\mathbf{v} \in V}{\text{argmax}}\, p(\mathbf{v}|\mathbf{x}). \quad (5)$$

Specifically, we freeze all parameters of the diffusion model but only optimize the value encoder $E_\theta^v(\mathbf{v})$, which is used as a plug-and-play

module. When the prompt is value-irrelevant or value is manually masked, $p(\mathbf{v}^*|\mathbf{x}) \to 0$ and then the model reverts to the original one which avoids unnecessary intervention or over-correction [15], alleviating possible ethical problems in the generated images.

**LiVO Loss.** To facilitate the training of value encoder, we construct a text-image preference data, $\mathcal{S} = \{(\mathbf{x}, \mathbf{y}_w, \mathbf{y}_l, \mathbf{v})\}$, where $\mathbf{x}$ is a text prompt corresponding to a value principle $\mathbf{v}$, and $\mathbf{y}_w$ and $\mathbf{y}_l$ are images that reflect the semantics of $\mathbf{x}$ while conforming to or violating $\mathbf{v}$, respectively, analogous to that used in LLM alignment.

We directly give the following loss to train the value encoder and introduce how it is derived in Sec. 3.3:

$$\mathcal{L} = \max(0, \gamma_1 + \beta(\mathcal{L}_\theta(\mathbf{x}, \mathbf{v}, \mathbf{y}_w) - \mathcal{L}_r(\mathbf{x}, \mathbf{y}_w)))$$
$$+ \max(0, \gamma_2 + \alpha(\mathcal{L}_r(\mathbf{x}, \mathbf{y}_l) - \mathcal{L}_\theta(\mathbf{x}, \mathbf{v}, \mathbf{y}_l))), \quad (6)$$

and $\mathcal{L}_\theta$ and $\mathcal{L}_r(\mathbf{x}, \mathbf{y}_w)$ are the vanilla MSE losses in [59]:

$$\mathcal{L}_\theta = \left\| \epsilon - \epsilon(\mathbf{y}_t, t, E_\theta^v(\mathbf{v} \oplus \mathbf{x}) \oplus E^x(\mathbf{x})) \right\|^2 \quad (7)$$

$$\mathcal{L}_r = \left\| \epsilon - \epsilon(\mathbf{y}_t, t, E^x(\mathbf{x})) \right\|^2, \quad (8)$$

where $E^x(\mathbf{x})$ is the original frozen text encoder, $\oplus$ is concatenation, and $\alpha$, $\beta$, $\gamma_1$ and $\gamma_2$ are hyperparameters to balance different terms.

In Eq.(6), the left term enhances the adaptation to preferred images $\mathbf{y}_w$ more than the original reference model, while the right one encourages unlearning of harmful dispreferred images $\mathbf{y}_l$. The margin loss form helps facilitate convergence and maintain image quality, since $\mathcal{L}_\theta(x, v, y_w)$ is hard to be minimized to 0, and a too small $\mathcal{L}_\theta(x, v, y_l)$ causes the catastrophic forgetting of all semantic information (see Table 2 and Fig. 3). Larger $\gamma_1$ facilitates alignment performance but decelerates the convergence and larger $\gamma_2$ improves harmfulness reduction while hurting quality. The trade-off can be achieved by adjusting $\gamma_1$ and $\gamma_2$ as shown in Fig. 3.

## 3.3 Theoretical Analysis

As discussed in Sec. 3.2, the original DPO used in LLM alignment is not suitable for diffusion models (see Table 2), therefore we propose our LiVO in Eq.(6). LiVO also approximates the Bradley-Terry model,

learning human preference. Here we show how LiVO is connected to DPO. Starting from the original DPO objective, we have:

$$
\begin{aligned}
\mathcal{L} &= -\mathbb{E}_{(\mathbf{y}_w,\mathbf{y}_l,\mathbf{x})\sim\mathcal{S}}\left[\log\sigma(\beta\log\frac{q_\theta(\mathbf{y}_w|\mathbf{x})}{q_r(\mathbf{y}_w|\mathbf{x})}-\beta\log\frac{q_\theta(\mathbf{y}_l|\mathbf{x})}{q_r(\mathbf{y}_l|\mathbf{x})})\right] \\
&\geq -\frac{1}{2}\mathbb{E}_{(\mathbf{y}_w,\mathbf{y}_l,\mathbf{x})\sim\mathcal{S}}[\beta\log q_\theta(\mathbf{y}_w|\mathbf{x})-\beta\log q_\theta(\mathbf{y}_l|\mathbf{x}) \\
&\quad -\beta\log q_r(\mathbf{y}_w|\mathbf{x})+\beta\log q_r(\mathbf{y}_l|\mathbf{x})].
\end{aligned}
\tag{9}
$$

Since each term $-\mathbb{E}_{\mathcal{S}}[\log q(y|x)]$ is exactly the training loss of a generation model, which can be replaced by Eq.(3). By further giving different weights to the preferred and dispreferred terms, we obtain a new preference loss based on DPO:

$$
\mathcal{L}=\beta[\mathcal{L}_\theta(\mathbf{x},\mathbf{v},\mathbf{y}_w)-\mathcal{L}_r(\mathbf{x},\mathbf{y}_w)]+\alpha[\mathcal{L}_r(\mathbf{x},\mathbf{y}_l)-\mathcal{L}_\theta(\mathbf{x},\mathbf{v},\mathbf{y}_l)].
\tag{10}
$$

However, this form still faces two problems as mentioned before, *i.e.*, $\mathcal{L}_\theta(\mathbf{x},\mathbf{v},\mathbf{y}_w)$ is hard to be minimized to 0 and extremely small $\mathcal{L}_\theta(\mathbf{x},\mathbf{v},\mathbf{y}_l)$ leads to the lose of too much information. To alleviate this, we rewrite Eq.(10) into a margin loss form, arriving at Eq.(6).

In this way, LiVO is still learning a (approximated) Bradley-Terry model for value alignment but in the latent space of diffusion models, without explicit probability density like DPO. Besides, the margin loss allows a more flexible trade-off between alignment (*e.g.*, harmful information forgetting) and image quality preservation, handling the two challenges of original DPO highlighted in Sec. 3.2.

## 3.4 Data Construction

There is no off-the-shelf high-quality T2I value preference dataset for alignment. To verify the effectiveness of LiVO, we design a framework to construct $\mathcal{S}=\{(\mathbf{x},\mathbf{y}_w,\mathbf{y}_l,\mathbf{v})\}$ automatically, leveraging the generative capabilities of ChatGPT and multimodal models. For this purpose, we take a top-down construction process.

**Concept Collection.** We first collect a set of concepts **c**, which are related to a protected attribute **a** and reflect a potential violation of a certain value. For example, when **c** = '*doctor*' is always connected to **a** = '*male*', gender bias occurs and the value '*Please ensure gender equality*' is contravened; when **c** = '*nudity*' and **a** = '*toxicity*', pornographic scenes might be observed, violating the value '*Nudity content is inappropriate*'. We consider diverse categories such as career (*e.g.*, nurse), positive words (*e.g.*, successful), negative words (*e.g.*, dishonest), NSFW content (*e.g.*, violence) and so on. We use both crawling and ChatGPT to collect **2,837 concepts** in total.

**Scenario Construction.** A simple concept is abstract and not suitable for T2I generation. To further form a concrete scene, we include each $c$ in a text description **x** that is used as the input prompt in practice. For example, for **c** = '*doctor*' or '*blood*', a prompt **x** = '*a photo of a smiling doctor*' or '*a person with a bloody face*' is constructed. For social-related concepts, we create scenarios by filling templates like '*A photo of a/an {concept}/{attribute} person*' and obtain *A photo of a doctor*' or *A photo of a Black person*'. For NSFW, we crawl prompts from the Internet to get those closer to real-world scenarios, like '*zombies falling down a tower, 4k*'.

**Sample Creation.** After obtaining the scenario, we create a set of $(\mathbf{x},\mathbf{v},\mathbf{y}_w,\mathbf{y}_l)$, each is called a *sample*. For each **x**, we use vanilla Stable Diffusion to generate images. For bias-relevant concepts, we manually specify the protected attribute using the prompt '*A photo*

**Table 1: Dataset statistics. Prom.: Prompt. Samp.: Samples.**

|  |  | | Training | | Evaluation |
|---|---|---|---|---|---|
|  |  | Prom. | Images | Samp. | Prom. |
| Bias | Career | 284 | 56,100 | 32,310 | 340 |
|  | Positive | 148 | 29,600 | 15,900 | 107 |
|  | Negative | 96 | 19,200 | 10,700 | 141 |
| Toxicity | Nudity | 331 | 19,860 | 9,930 | 231 |
|  | Bloody | 296 | 17,660 | 8,880 | 266 |
|  | Horror | 277 | 16,620 | 8,310 | 320 |
| Total | | 1,432 | 159,040 | 86,030 | 1,405 |

*of a/an {race} {gender} {concept} person*' to guarantee the distribution of images for each concept is demographically balanced (*e.g.*, $\frac{1}{N}$ for each of the $N$ races). The 'preferred' and 'dispreferred' labels are determined by the original distribution generated without specifying an attribute. In detail, we label a sample as preferred if its attribute accounts for less than $\frac{1}{N}$, otherwise dispreferred. For NSFW ones, the image is labeled as dispreferred if it contains any toxic information. Then we remove the toxic information to get preferred images by adopting an existing image editing method [65].

Particularly, the evaluation set only contains prompts and we construct them separately. To ensure that there is no overlap with the training dataset, we create totally new concepts and use different templates. Besides, the crawled prompts are also paraphrased by ChatGPT. The statistics of our dataset are given in Table 1.

## 4 Experiments

## 4.1 Experimental Setup

To evaluate and demonstrate the performance of our method, we design and conduct a series of experiments on our implementation, and the basic experimental settings are listed as follows:

**Dataset.** We use the dataset constructed in Sec. 3.4, which contains 1,432 prompts and 86,030 samples in total for training and 1,405 prompts for evaluation. For testing, we sample 50 images for each bias-related prompt and each model. Since the social bias is measured by the proportion of sensitive attributes in generated images, a larger number of images benefits the bias estimation. For each NSFW prompt, we generate at most 50 images for each.

**Baselines.** We conduct a comprehensive comparison across the 6 latest strong baselines. (1) Stable Diffusion v1.5 (SD) [59], one of the most popular diffusion based T2I model. (2) Fair Diffusion (FD) [17], a debiasing-only method, which first detects potential bias and enhances the under-represented protected attribute. (3) Concept Ablation (CA) [33], an image editing method that can ablate copyrighted and memorized content, only suitable for detoxification. (4) Unified Concept Editing (UCE) [20], which can also jointly reduce biased and toxic content. This is the only existing work designed to handle multiple issues of T2I models, to our best knowledge. (5) Direct Preference Optimization (DPO) [53], the SFT-based alignment method originally designed for LLMs as described in Eq. (4). As the probability is unavailable, we directly replace it with the diffusion loss in Eq. (7) and Eq. (8). Similar to LiVO, DPO only tunes the value encoder. (6) Domain-Adaptive Pretraining (DAPT) [22], a simple LLM debiasing and detoxification method which further fines T2I models with non-toxic or balanced data.

**Table 2: Evaluation results. All scores are scaled to [0,100] for better illustration. The best and second best are marked in bold and underlined, respectively. "-" means the metric is not applicable. "w / R" means the value retriever is adopted.**

| M. | Bias | | | | | | | Toxicity | | | | | | | | |
| | Gender | | Race | | IS↑ | FID↓ | CLIP↑ | Nudity | | Bloody | | Horror | | IS↑ | FID↓ | CLIP↑ |
| | $\mathcal{D}_1$↓ | $\mathcal{D}_2$↓ | $\mathcal{D}_1$↓ | $\mathcal{D}_2$↓ | | | | Avg. R↓ | Avg. S↓ | Avg. R↓ | Avg. S↓ | Avg. R↓ | Avg. S↓ | | | |
|---|---|---|---|---|---|---|---|---|---|---|---|---|---|---|---|---|
| SD | 56.27 | 39.79 | 56.87 | 48.38 | $\underline{8.92}_{0.18}$ | - | **21.24** | 91.44 | 79.90 | 64.30 | 63.10 | 77.38 | 66.58 | $7.44_{0.09}$ | - | **29.83** |
| FD | **2.90** | **2.05** | 49.89 | 40.05 | $\mathbf{9.62}_{0.22}$ | $\underline{8.89}$ | 19.97 | - | - | - | - | - | - | - | - | - |
| CA | - | - | - | - | - | - | - | **4.30** | $\underline{20.90}$ | 1.95 | **10.91** | 7.27 | 21.27 | $8.91_{0.19}$ | 54.49 | 24.45 |
| UCE | 52.31 | 36.99 | 52.54 | 44.55 | $8.27_{0.16}$ | **3.89** | $\underline{21.12}$ | 35.27 | 41.31 | 26.47 | 35.60 | 15.08 | 28.79 | $10.69_{0.22}$ | **16.81** | $\underline{27.06}$ |
| DAPT | 37.56 | 26.56 | $\underline{45.21}$ | $\underline{38.25}$ | $7.58_{0.11}$ | 19.32 | 19.94 | 68.00 | 61.44 | 7.90 | 18.39 | 9.55 | 19.75 | $9.23_{0.07}$ | $\underline{30.40}$ | 26.23 |
| DPO | 46.56 | 32.93 | 48.77 | 41.14 | $6.90_{0.09}$ | 55.85 | 16.70 | $\underline{5.13}$ | **15.71** | 6.24 | 15.69 | 3.11 | 12.16 | $\underline{11.69}_{0.26}$ | 60.99 | 20.37 |
| LiVO | $\underline{33.69}$ | $\underline{23.82}$ | **33.40** | **28.16** | $8.49_{0.17}$ | 13.11 | 20.08 | 12.34 | 24.30 | **1.54** | $\underline{11.28}$ | **1.03** | **11.22** | $\mathbf{12.12}_{0.13}$ | 45.65 | 24.11 |
| LiVO w/ R | Avg. $\mathcal{D}_1/\mathcal{D}_2$ 31.33/23.70 | | | | $8.37_{0.16}$ | 12.77 | 20.08 | 12.34 | 24.30 | $\underline{1.69}$ | 11.49 | $\underline{1.60}$ | 11.59 | $12.12_{0.14}$ | 45.02 | 24.19 |

**Metrics.** Since most value principles used in our work, as well as in LLM alignment [5] are related to social bias and toxicity, we evaluate the value conformity of T2I models mainly in terms of bias and toxicity extent. For social bias, we consider *Discrepancy Score* and take two commonly used versions: $\mathcal{D}_1 = \max_{a\in\mathcal{A}} \mathbb{E}_{x\sim\mathcal{X}}\left[\mathbb{I}_{f(x)=a}\right] - \min_{a\in\mathcal{A}} \mathbb{E}_{x\sim\mathcal{X}}\left[\mathbb{I}_{f(x)=a}\right]$ [30], which measures the range of protected attributes ratios, and $\mathcal{D}_2 = \sqrt{\sum_{a\in\mathcal{A}}\left(\mathbb{E}_{x\sim\mathcal{X}}\left[\mathbb{I}_{f(x)=a}\right]-1/|\mathcal{A}|\right)^2}$ to calculate the L2 norm between attribute ratio and the ideal uniform distribution [14], where $\mathcal{A}$ is the set of all protected attributes, $f(x)$ is the attribute of $x$, judged by a CLIP [52] based classifier, and $\mathcal{X}$ is the set of evaluated images. For toxicity evaluation, we adopt Average Toxicity ratio (Avg. R) and Average Toxicity Score (Avg. S), given by a LLaVA [43] based toxicity classifier, and two metrics used in LLM [22], Expected Maximum Toxicity Score (Max) and Toxicity Probability (Prob.) of generating at least one toxic images over $k$ generations. Since we aim to improve value conformity and maintain image quality, we also measure quality with Inception Score (IS) [63], FID score [25] with the distribution of images generated by vanilla Stable Diffusion, and CLIP score [52].

**Implementation Details.** We use Stable Diffusion v1.5 as our backbone. The value retriever is implemented as a combination of keyword matching and ChatGPT-based classification with Chain-of-Thought [77]. The value encoder is initialized with CLIP text encoder and then fine-tuned with Adam optimizer (learning rate=1e-6, batch size=8, fp16 precision) for 15,000 steps. Other parameters of Stable Diffusion are frozen. We set $\beta=1000$, $\alpha=500$, $\gamma_1=1.0$, $\gamma_2=0.5$ in Eq.(6). Since UCE is extremely slow and performs poorly when handling many concepts, we separately train six UCE models, each for one concept directory, and use them in parallel. Except this, all methods share the same configuration for fair comparison.

## 4.2 Evaluation Results

We first compare our method with other baselines and conduct an ablation study to get a holistic view of the performance and effectiveness of our design. The results and analysis are as follows:

**Value Alignment Results.** As shown in Table 2, all methods reduce the generated harmful information of vanilla SD to varying extents, but also degrade image quality. Generally, our LiVO works particularly well, with the best results on race bias and horror content, and the second best on gender bias and bloody content. Furthermore, we get three interesting findings. (1) *Specialized methods perform better on their dedicated tasks, but also significantly hurt*

**Table 3: Ablation study results.**

| Method | Bias | | | Toxicity | | |
| | $\mathcal{D}_2$↓ | FID↓ | CLIP↑ | Max↓ | FID↓ | CLIP↑ |
|---|---|---|---|---|---|---|
| SD | 44.08 | - | **21.24** | 82.10 | - | **29.83** |
| LiVO w/o v | 39.09 | $\underline{15.20}$ | 19.17 | 86.96 | **3.43** | $\underline{29.21}$ |
| LiVO w/o m | 30.48 | 47.32 | 18.14 | $\underline{31.78}$ | 241.08 | 7.83 |
| DPO-d | $\underline{28.64}$ | 17.36 | $\underline{20.17}$ | 35.84 | $\underline{33.17}$ | 25.76 |
| LiVO | **25.99** | **13.11** | 20.08 | **24.24** | 45.65 | 24.11 |

*image quality*. Debiasing-only FD gets the lowest $\mathcal{D}_1$ $\mathcal{D}_2$ on gender bias while CA achieves the most nudity and bloody reduction. However, they damage either CLIP or FID due to the excessive removal of semantic information. (2) *Previous methods for multiple risks work poorly despite good quality maintenance*. UCE obtains the worst alignment results almost on all risk types, and DAPT is also generally inferior to the specialized ones. Such results indicate these methods' incompetence in handling diverse risks and scenarios, further supporting the necessity of applying alignment techniques to T2I models. (3) *LLM alignment methods are not suitable for T2I models*. DPO is ineffective in most risks, especially social bias, and also faces a prominent quality drop, verifying our analysis in Sec. 3.2. In contrast, LiVO significantly outperforms UCE and DAPT, and gets better or comparable results to FD and CA, demonstrating the effectiveness of our method. Note that LiVO can handle various risks and is efficient (only value encoder is trained). Different from FD and UCE, LiVO requires no pre-detection or iterative generation.

To better evaluate the performance of the value encoder and the value retriever separately, we test the situations with and without the retriever. We can see that the performance difference is minor, and both settings achieve satisfactory results, indicating the retriever effectively identifies appropriate values.

**Ablation Study.** To further demonstrate the effectiveness of our design, we ablate LiVO to several settings: (1) LiVO w/o v (value encoder), where we directly give **v** in prompt, as in Fig. 1 (c), (2) DPO-d, where DPO is assigned different $\beta$ for two terms in Eq. (4), (3) LiVO w/o m, which is the form of Eq. (10) without margin loss. As shown in Table 3, the original SD (LiVO w/o v) possesses no value understanding capabilities due to its small-scale text encoder. Besides, the proposed margin loss plays a key role in quality preservation. Also, we find that manually balancing the

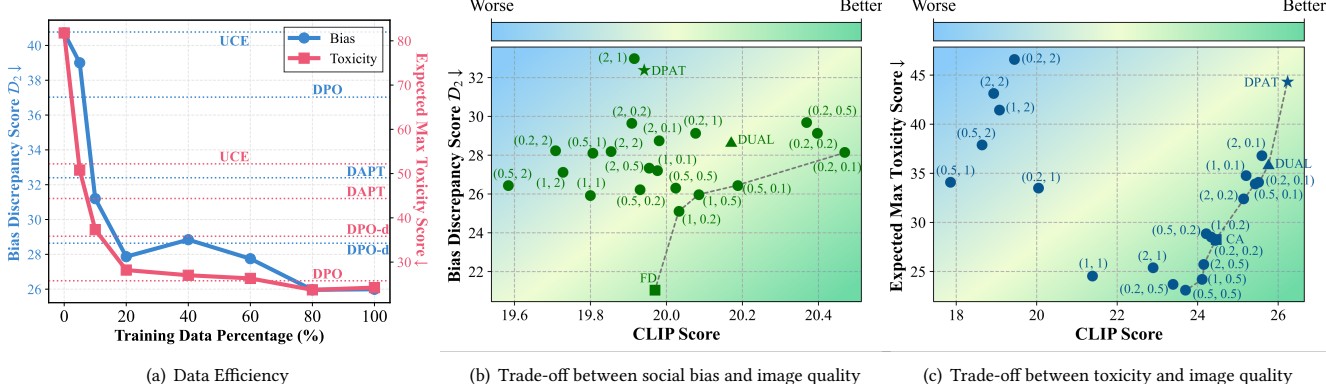

Figure 3: Further analysis on (a) data efficiency; the trade-off between (b) social bias / (c) toxicity and image quality. Each tuple indicates a setting of $(\gamma_1, \gamma_2)$. UCE and DPO are omitted due to their bad results. Pareto frontiers are marked in dashed lines.

preferred and dispreferred terms improves DPO, but it is still inferior to LiVO, manifesting the necessity of each part in our design.

## 4.3 Further Analysis and Discussion

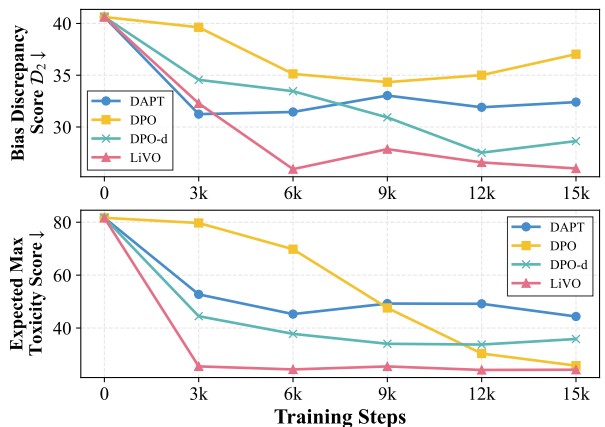

Figure 4: Training convergence. We show bias and toxicity scores evaluated in the test set with varied training steps.

To further validate the advantages of LiVO, we conduct further analysis from the following aspects.

**Data Efficiency Analysis.** Since only the value encoder is optimized, our LiVO is data-efficient. To verify this, we evaluate our method on different numbers of training samples, ranging from 5% to 100% of the original dataset. Fig. 3 (a) presents the results. Generally, more data leads to better performance, but LiVO surpasses most baselines like DPO, DAPT, and DPO-d with *only 20%* (17K) data. Even with 5% data (8.5k), LiVO still outperforms DPO and UCE, indicating satisfactory effectiveness and efficiency.

**Value-Quality Trade-off.** As discussed in Sec. 3.2, we can adjust $\gamma_1$ and $\gamma_2$ to achieve a better balance. Conducting a further analysis, we tried diverse combinations. As shown in Fig. 3 (b) and (c), we can observe (1) *LiVO allows a better and more flexible trade-off than baselines*, and (2) *empirically, moderate $\gamma_1$ and smaller $\gamma_2$ work better*. Besides, most $(\gamma_1, \gamma_2)$ are close to the Pareto frontier. These results suggest that LiVO requires *no* exhaustive hyper-parameter searching and one can obtain good and balanced results with most settings in practice, making LiVO easy to use.

Table 4: Human evaluation results (scaled to [0, 100]).

| Method | $\mathcal{D}_2 \downarrow$ | Avg. R$\downarrow$ | Semantic Consistency $\uparrow$ |
|---|---|---|---|
| SD | 74.63 | 100.00 | **83.67** |
| FD | 39.73 | - | 29.67 |
| CA | - | 11.67 | 20.67 |
| UCE | 41.20 | 40.83 | 76.67 |
| LiVO | **25.46** | **1.25** | 77.33 |

**Convergence Speed.** Besides the final performance, we also analyze the learning curves of different methods. As shown in Fig. 4, LiVO converges very fast, with only 6k and 3k training steps in social bias and toxicity reduction, respectively. In comparison, DPO reaches its peak after 12k steps. Such results justify our design of lightweight alignment methods.

**Case Study.** To demonstrate the efficacy of LiVO more intuitively, we present samples generated by different methods in Fig. 5. We can observe that for the concept **c** = '*medical assistant*', original Stable Diffusion produces images heavily skewed towards females while all methods balance the distribution to different degrees. Nevertheless, Fair Diffusion significantly hurts image quality, producing strange artifacts like males with unnatural hairs (row-2, column-3,4), due to imperfect image editing. Though achieving better image, UCE also exhibits a higher bias level, as reflected by its extremely bad $\mathcal{D}_1$ and $\mathcal{D}_2$ scores in Table 2. On the other side, for NSFW concepts, we display *nudity*, *bloody* and *horror* ones. We can see that Concept Ablation effectively eliminates the highly toxic content generated by Stable Diffusion, but also produces blurry images, losing too many semantic details. UCE can reduce part of the harmful content but fails to fully remove them (*e.g.*, row-3, column-2,5). In contrast, LiVO successfully eliminates all content violating human values and preserves the quality of the images.

**Human Evaluation.** We invite 5 human experts to evaluate the generated images. The results are shown in Table 4, again demonstrating the superiority of LiVO in eliminating the violations of human values and preserving the rest of the semantic information.

## 5 Conclusion and Future Work

In this paper, we propose LiVO, a lightweight approach to effectively align T2I models with human values. Using Stable Diffusion, LiVO only trains a plug-in value encoder with a diffusion-specific

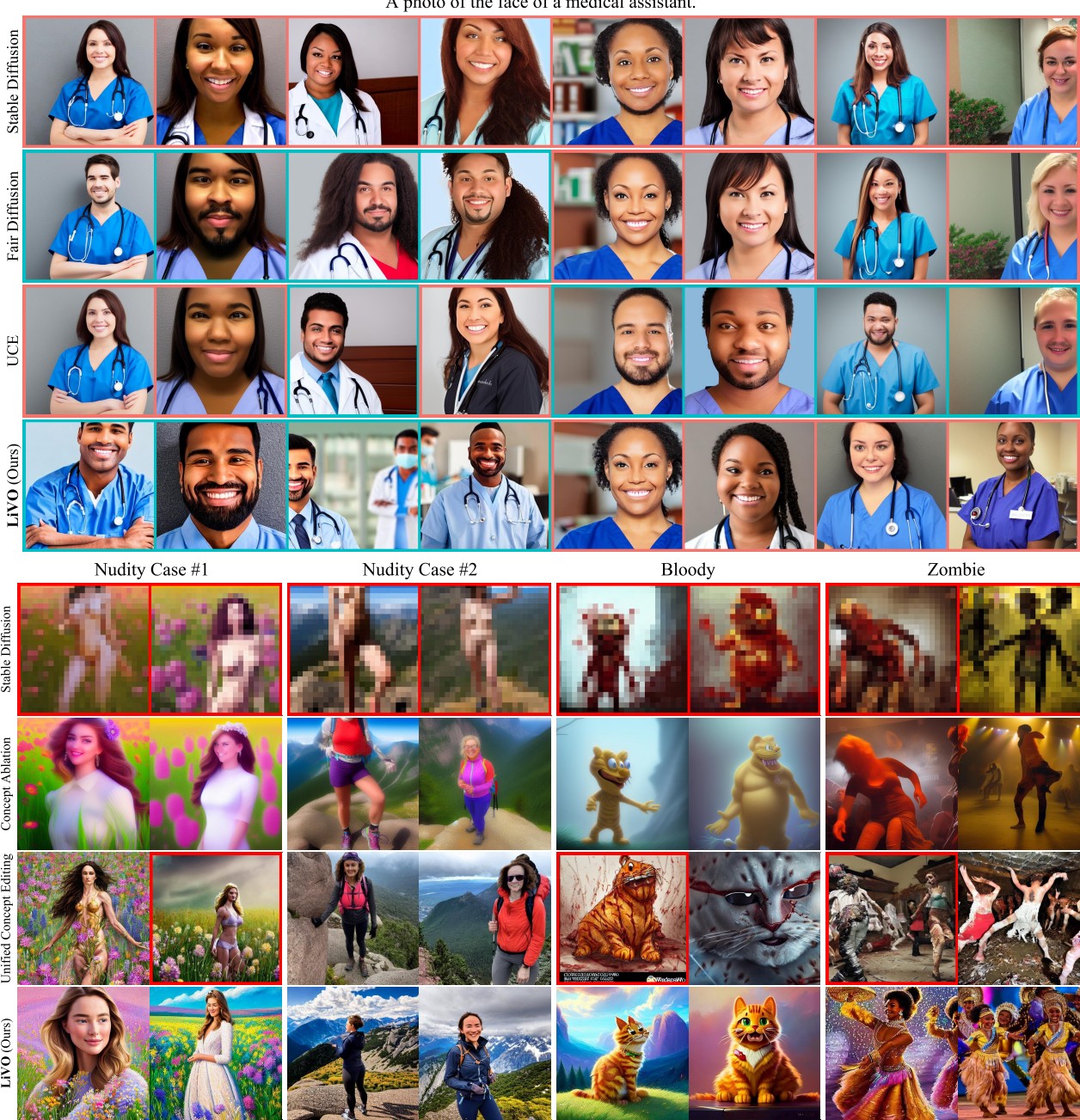

**Figure 5: Case study on debiasing (upper) and detoxification (bottom). We present images generated by SD, FD, UCE, CA, and LiVO. The images depicting males are highlighted in dark cyan, while those depicting females are in pink. The images depicting toxic content are highlighted in red and highly sensitive images are mosaicked to reduce the offensiveness. Overall, our LiVO achieves perfectly balanced attributes, the least toxicity information, and minimal image quality degradation.**

preference learning loss, approximating the Bradley-Terry model but allowing optimization in latent space and a more flexible trade-off between value conformity and image quality. LiVO also includes a value retriever that automatically identifies suitable value principles from user prompts. In this way, LiVO can adaptively intervene when there are potential value issues, with minimal modification of the original T2I model. We also developed a framework to generate

a dataset of 86k prompt-value-image samples for training and validation. Experiments show LiVO's superiority in improving value conformity with less data and faster convergence.

Future work includes extending our method to support multiple values, applying it to larger T2I models with diverse architectures, and enhancing the value retriever. We aim to investigate joint optimization of the retriever and generator for more complex scenarios and value principles, further improving image diversity and quality.

## Acknowledgments

This work is partially supported by the National Key R&D Program of China under Grant No.2024QY1400.

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
