# OpenReview forum: "Embedding an Ethical Mind:  Aligning Text-to-Image Synthesis via Lightweight Value Optimization"
_acmmm.org/ACMMM/2024/Conference — MM2024 Poster_

### Official Review · Reviewer_AjUw · 2024-05-24

**Rating:** 4
**Confidence:** 3

**Summary:**

This paper proposes LiVO (Lightweight Value Optimization), a novel lightweight method for aligning T2I diffusion models with human values to reduce generation of harmful content. LiVO optimizes a plug-and-play value encoder to integrate specified value principles with the input prompt, allowing control over both semantics and values of generated images. It uses a diffusion model-tailored preference optimization loss and an automatically constructed text-image preference dataset. Without updating most model parameters, LiVO significantly reduces harmful outputs while maintaining image quality by adaptively selecting values from the input prompt.

**Strengths:**

1. The content of this work is particularly needed and interesting in the community at present.
2. This method is a good job for ethically aligned T2I models.
3. The author's supporting material explains the author's approach in detail.

**Limitations:**

1.	Fig. 1 is very strange. It shows that LLM is less affected by bias, giving the reader a sense that LLM is used to solve sensitive problems in the following solution. However, the LLM does not appear to be directly involved in the image generation part, which creates a misleading impression. It would be better to change or remove Fig.1(c).
2.	The author's description of the value retriever is very vague. It is necessary to specify how to retrieve and classify those value prompts in the value retriever.
3.	I wonder why the author needs a separate value encoder instead of fine-tuning the text encoder.
4.	Experiments show that this method seems to have poor performance in semantic consistency (CLIP SIM). I think additional experiments are needed to prove that semantic consistency decline only appears in those bias semantics, while other semantics are consistent.
5.	Will the proposed dataset be open source in the future?

**Suitability:**

3

---

### Official Review · Reviewer_tUQu · 2024-05-24

**Rating:** 5
**Confidence:** 2

**Summary:**

The paper proposed a novel lightweight method LiVO to achieve Text-to-Image(T2I) model alignment with human values. The authors design a diffusion model-tailored preference optimization loss to make a trade-off between image quality and value conformity. The paper also builds a framework to automatically construct a text-image preference dataset of 86k samples. The results show that the proposed LiVO outperforms several baselines.

**Strengths:**

1. The research topic is novel and meaningful.
2. The theoretical analysis is convincing.
3. The cases shown in Figure 5 provide good support for the conclusion.

**Limitations:**

Though comprehensive experiments are conducted, it is better to include human evaluation to prove the effectiveness of the method.

**Suitability:**

3

---

### Official Review · Reviewer_SmPb · 2024-05-26

**Rating:** 3
**Confidence:** 3

**Summary:**

This paper proposes LiVO, a novel lightweight method to align T2I models with human value principles. Specifically, a value encoder, a value retriever and a LiVO Loss are proposed. Besides, a text-image value preference dataset is contrusted, comprising 86k (prompt, value-aligned image, value-violating image, value principle) samples.

**Strengths:**

1.	The LiVO loss is novel. It is a margin loss, derived from DPO. The authors pioneeringly extend DPO into the diffusion models to separate normal results and human value preferred results.
2.	The writing and representation are logical and easy to understand.

**Limitations:**

I have some concerns about the new dataset and experiments:
1.	In the new dataset, the images are generated through SD. The generation contains bias itself, and I don’t think these images could be ground truth for this task.
2.	It looks like the prompts in the test set and training set are separated, but the images may be overlapped.
3.	In the baseline, why not add [14] and [30].
4.	The ground truth is generated by SD. In Table 2, why the IS and Clip scores for SD is suboptimal?
5.	Besides experiments on the newly proposed dataset, why not use the experiment setting in [14]?

Typos probably:
Line 521, Avg.R should be Avg.S?

**Suitability:**

3

---

### Official Review · Reviewer_nqpT · 2024-06-03

**Rating:** 5
**Confidence:** 2

**Summary:**

This paper proposes a method called LiVO (Lightweight Value Optimization) to align Text-to-Image (T2I) models with human values. ​ LiVO optimizes a value encoder using a diffusion-specific preference learning loss, reducing harmful outputs and achieving faster convergence compared to baselines. ​ The key contributions of this work are in three folds - Firstly, LiVO utilizes a diffusion model-tailored preference optimization loss, which approximates the Bradley-Terry model commonly used in LLM alignment. ​ This loss function allows for a flexible trade-off between image quality and value conformity, providing a theoretical foundation for aligning T2I models with human values. Secondly, the development of a framework for constructing a text-image preference dataset, and finally demonstrating LiVO's effectiveness in reducing toxic content and improving human value conformity via comprehensive experiments across diverse risk types.

**Strengths:**

This paper has following strengths: ​

1. Novelty: The paper addresses the important and relatively unexplored challenge of aligning Text-to-Image (T2I) models with human values. The proposed LiVO method introduces a novel lightweight approach to tackle this challenge. ​

2. Adequate Evaluation: The paper presents comprehensive experiments and analyses to demonstrate the effectiveness of LiVO. ​ It compares LiVO against several strong baselines and shows that it significantly reduces harmful outputs while achieving faster convergence. ​ The evaluation includes a diverse text-image value preference dataset, covering various value misalignment scenarios.

3. Clarity: The paper is well-written and clearly presents the problem statement, the proposed method, and the experimental results. The authors provide detailed explanations of the LiVO framework, the value encoder optimization, and the construction of the text-image preference dataset. ​

4. Applications: The paper highlights the potential applications of value-aligned T2I models in various domains, such as creative arts, advertising, and education. By addressing the ethical concerns associated with T2I models, LiVO opens up possibilities for responsible and ethical use of these models in real-world applications.

**Limitations:**

Some of the limitations I would like the authors to comment on are as follows:

1. Generalizability: The paper does not extensively explore the generalizability of the LiVO method to different T2I models and datasets, which limits its effectiveness in different contexts. ​

2. Limited Scope of Value Principles: The choice of predefined value principles may not encompass the full range of ethical considerations or cultural variations.

3. Lack of Real-World Data: The evaluation of LiVO relies on a constructed dataset, which may not fully capture the complexity and diversity of real-world data. ​

**Suitability:**

3

---

### Meta-Review · Area_Chair_QBkf · 2024-07-04

**Recommendation:** Accept (Poster)
**Confidence:** 5

**Metareview:**

The paper presents a novel approach, LiVO, aimed at aligning Text-to-Image (T2I) models with human values. The method introduces a value encoder, a value retriever, and a LiVO loss, and constructs a text-image value preference dataset. LiVO is designed to reduce harmful outputs and achieve faster convergence compared to existing baselines. The approach is evaluated comprehensively through experiments, demonstrating its effectiveness in aligning T2I models with human values. Based on the reviews and the authors' rebuttal, the consensus leans towards acceptance. The paper addresses an important and novel challenge in T2I model alignment with human values, provides a comprehensive evaluation, and clearly articulates its contributions. However, future work should focus on enhancing the generalizability of the method, incorporating a wider range of value principles, and validating the approach with real-world data to address the raised concerns fully.